# VISTA: A Panoramic View of Neural Representations

**Tom White**
School of Design Innovation
Victoria University of Wellington
Wellington, New Zealand 6011
`tom.white@vuw.ac.nz`

## Abstract

We present VISTA (Visualization of Internal States and Their Associations), a novel pipeline for visually exploring and interpreting neural network representations. VISTA addresses the challenge of analyzing vast multidimensional spaces in modern machine learning models by mapping representations into a semantic 2D space. The resulting collages visually reveal patterns and relationships within internal representations. We demonstrate VISTA's utility by applying it to sparse autoencoder latents uncovering new properties and interpretations. We review the VISTA methodology, present findings from our case study[1], and discuss implications for neural network interpretability across various domains of machine learning.

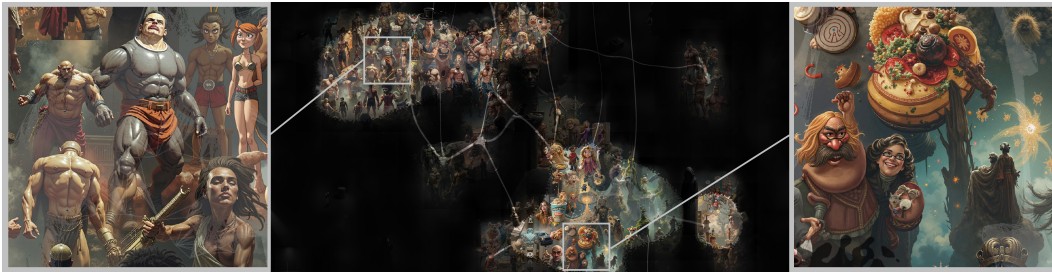

Figure 1: Activations of Gemma2-2B residual latent 20-9745 as a VISTA map aggregating thousands of inputs. The automated explanation given to this latent by Gemma Scope is "references to muscle-related subjects and terminology". Zooming into the upper cluster (far left) reveals a collection of muscle related themes. However examining the lower cluster (far right) reveals a collection of subclusters sharing only word morphology such as "mustache", "mushroom", and "mystical". (link)

## 1   Introduction

Deciphering the internal representations of neural networks and biological brains remains a formidable challenge in both artificial intelligence and neuroscience. As models grow in complexity and scale, traditional methods of analysis struggle to provide comprehensive insights into the vast landscape of learned features and concepts. This challenge is further compounded by the emerging evidence that different learning systems, when exposed to similar stimuli, often develop comparable internal representations – a phenomenon that demands novel approaches for investigation and understanding (Sucholutsky et al., 2023).

---

[1]`http://got.drib.net/latents/`

38th Conference on Neural Information Processing Systems (NeurIPS 2024).

Building on previous work focusing on visualisations for interpretability(Carter et al., 2019), we propose VISTA (Visualization of Internal States and Their Associations) which combines clustering with text-to-image models to create a visual "cartography" of neural representations. VISTA offers a unique approach to exploring and interpreting high-dimensional spaces by generating visual collages that map the semantic relationships between different elements of a representation. This method aims to provide researchers with an intuitive, visual interface for navigating the complex terrain of neural representations, potentially revealing patterns and structures that might otherwise remain hidden.

As an initial test of VISTA's capabilities, we apply our method to the domain of the latent codes of sparse autoencoders (Cunningham et al., 2023; Bricken et al., 2023), an area where the current state of the art relies heavily on automated interpretability techniques using large language models (LLMs) (Bills et al., 2023). We examine three use cases using the Gemma-2B SAE model (Lieberum et al., 2024), comparing VISTA's findings with those obtained through automated methods. Our results indicate that while VISTA often corroborates the features identified by LLM-based techniques, it also demonstrates the ability to uncover deeper meanings and associations that automated methods currently miss. This suggests that VISTA could serve as a valuable complementary tool in the broader effort to understand and unify representations across different neural models.

## 2   Methodology

The VISTA pipeline consists of several steps designed to create an interactive, visual representation of high-dimensional data.

1. **Representation and Dataset Selection:** Choose a dataset of interest and a representation that assigns a feature vector to each input.

2. **Data Encoding:** Encode the dataset using the selected representation.

3. **Dimensionality Reduction:** Apply UMAP clustering to project the encoded data into a 2D space, preserving nearest neighbor relationships from the higher-dimensional representation (McInnes et al., 2020).

4. **Cartographic Rendering:** Generate a visual map of the 2D space. This process involves dividing the 2D space into smaller tiles, extracting the subset of the original dataset that corresponds to each area of the UMAP projection, and rendering one item from each subset as part of a MultiDiffusion panorama (Bar-Tal et al., 2023).

5. **Interactive Visualization:** Present the resulting graphic in an interactive interface, allowing exploration of the visual space alongside the corresponding data points.

The cartographic rendering has been iteratively refined to highlight semantic structure. The UMAP clusters abstracted into areas of high probability with connections between clusters explicitly shown and overlaid with semantic information from the text-to-image process. Taking advantage of the iterative nature of the diffusion pipeline, we select different inputs within each region at each step of the diffusion process to semantically smooth out the drawing process and remove outliers. Together these produce a more coherent and representative visualization.

The final result is an interactive graphic that invites exploration. Visual anomalies naturally draw attention, encouraging closer inspection and potentially revealing unexpected patterns or relationships within the data.

Our efforts of understanding shared representational structure are grounded in using mutual nearest neighbor metric for measuring alignment. This is consistent with recent research on representational alignment (Huh et al., 2024). The fidelity of the resulting VISTA map itself can also be measured directly as nearest neighbor analysis is meaningful across both low and high dimensional spaces. We report these map accuracies as mutual-knn "gain" which explicitly removes the null hypothesis of chance neighbor pairings as k increases (see appendix A for notes).

In the following section, we present a specific use case and implementation results to demonstrate the effectiveness of the VISTA pipeline in practice.

# 3 Experiment: Interpreting Sparse Latents

Sparse AutoEncoders (SAEs) have emerged as a promising technique in machine interpretability, but understanding the "meaning" encoded in each of the thousands of latents remains elusive. Current best practice is "Automated Interpretability" in which the latents are fed into LLMs and they come up with suggested labels for each of the latents (Bills et al., 2023).

We constructed a VISTA pipeline to explore the latents of the publicly released Gemma2-2B SAE latents (Lieberum et al., 2024). Here we use the implementation covered above and look at the resulting visualizations for 3 latents in particular to see how our approach is compatible with the automated techniques, but able to surface new and previously hidden properties.

## 3.1 Implementation

Following the steps outlined in our methodology:

- **Representation and Dataset:** We chose Gemma2-2B SAE 16k residual layer 20 as the representation. For the dataset we chose 200,000 auto-generated short captions from the Human Preference Synthetic Dataset (ProGamerGov, 2024) which follows the methodology of Betker et al. (2023). For each map we take the top 4000 (2%) captions that maximize the selected latent, and each caption is represented by this SAE representation.

- **Dimensionality Reduction:** We applied a custom distant metric which sums the cosine distance in SAE space with the absolute difference along the latent axis. This was mapped into a UMAP with an asymmetric aspect ratio.

- **Cartographic Rendering:** We used a version of MultiDiffusion running on the Flux.1 [dev] diffusion model (Labs, 2024), with 100 diffusion steps and a minimum of 4 points per cluster for semantic smoothing.

- **Interactive Visualization:** The resulting graphic was placed in an interactive interface for exploration, with the original 4000 captions arranged spatially to preserve nearest neighbors.

These choices provided a practical way to make an initial evaluation of our approach. The dataset was selected as it contains a variety of subjects and is already well suited for use in a text-to-image context. The UMAP hyperparameters were intended to allow "stretching" along the latent axis and though this wasn't always achieved as intended we did see stable clustering across runs with different random seeds. Using a single A100 GPU we could generate one 144 megapixel (9k x 16k) panorama image in a few hours. VISTA interfaces for twelve Gemma Scope latents were created and can be explored in our online interface; three of these are explored below as case studies.

## 3.2 Case Studies

We examined three specific latents to demonstrate the capabilities of our VISTA approach:

### 3.2.1 Case Study 1: "ingredients" [gemma-2-2b/20-res-16k/5011]

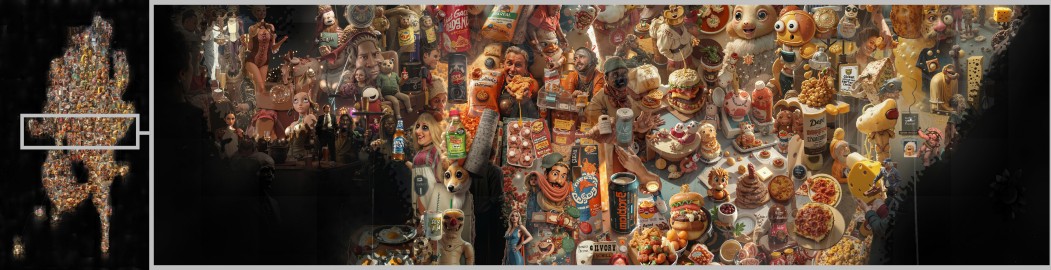

Figure 2: VISTA map for latent 20-5011 (left) with detail view (right). (link).

Gemma Scope label: "ingredients and dishes related to food preparation and recipes" (link)

Our findings: Visual inspection confirmed a large number of food-related ingredients, supporting the Gemma Scope label. The visualization hinted at specific ingredients triggering this latent code and their proportions in the dataset. For example, Figure 2 includes inputs referencing champagne, beer, iced tea, chips, honey, pizza, pretzels, and cheese. This map has a mutual-knn gain max of 0.129 at k=9%.

### 3.2.2  Case Study 2: "muscle" [gemma-2-2b/20-res-16k/9745]

Gemma Scope label: "references to muscle-related subjects and terminology" (link)

Our findings (see Figure 1): Initial inspection agreed with the label, showing many regions with muscular imagery. However, we discovered an additional pattern: about 40% of the inputs lacked muscular references, instead clustering around words beginning with "M" (e.g., "musical", "mustache", "mystic", "mossy"). This suggests the latent is also weakly triggered by certain "M" words. This map has a stronger mutual-knn gain max of 0.273 at k=5%.

### 3.2.3  Case Study 3: "indebted" [gemma-2-2b/20-res-16k/9220]

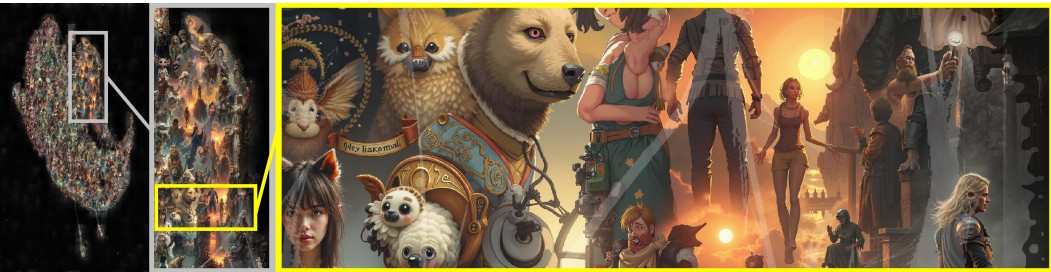

Figure 3: VISTA map for latent 20-9220 (left) shows high level patterns such as the stripes in the detail view (center). Zooming in further (right) we discover distinct clusters of animal combinations ("resembling a bear or big"), sky combinations ("looking at a sunset or sunrise") and style combinations ("a medieval or fantasy setting"). (link)

Gemma Scope label: "expressions related to legal or financial obligations" (link)

Our findings: This case was the most divergent from the Gemma Scope label. VISTA visualization showed clear visual clusters with no references to finance. Further examination revealed that this latent is triggered by conjunctions of visual forms, such as "a dramatic sunset or sunrise", "in a medieval or fantasy setting", "in a black & red attire", and "a unique half-deer, half-human figure". This discrepancy could be due to domain skew between datasets or limitations in automated techniques for identifying complex associative patterns. This map has a relatively weaker mutual-knn gain max of 0.099 at k=12%.

These case studies demonstrate VISTA's ability to confirm automated interpretations and uncover additional, sometimes unexpected, patterns in SAE latents.

## 4   Conclusion

VISTA (Visualization of Internal States and Their Associations) introduces a novel approach to exploring and interpreting neural representations through visual cartography. Our experiment with sparse latents from the Gemma2-2B model demonstrates VISTA's potential to complement and extend current automated interpretability techniques.

Key findings include:

- Confirmation of some automated interpretations, validating VISTA's basic functionality.
- Discovery of additional patterns not captured by automated methods, such as secondary triggers for certain latents.
- Revelation of unexpected associations, highlighting potential limitations in current automated techniques.

These results suggest that visual, interactive exploration can provide valuable insights into complex neural representations. VISTA offers a promising direction for future research in AI interpretability, potentially bridging the gap between automated analysis and human intuition.

As we continue to develop and refine VISTA, we anticipate its application to a broader range of neural representations and its integration with existing interpretability pipelines. This work represents an early step towards more comprehensive and intuitive understanding of neural networks, contributing to the broader goal of unifying representations across different models.

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

## A   Notes on mutual knn gain metric

We have attempted to quantitatively measure the fidelity of the resulting VISTA visualizations using mutual-knn scoring as is covered in more detail in (Huh et al., 2024). We have found two conventions useful in adapting this metric:

1) We generally express k as a percentage instead of an absolute number, which is easier to compare across different size datasets. We have also found in practice that mutual-knn can be quickly estimated my taking a smaller subset of the data, which is useful for iterative development. Each of our case study datasets were 4000 points, so absolute k reported can be found by multiplying (eg: mutual-knn at 1% refers here to k=40)

2) We generally allow higher values of k (5-10%) than usually used found in alignment studies as we expect mapping down to two dimensions to be quite lossy. However for unaligned and randomly distributed data, the mutual-knn will grow linearly with k. So we report mutual knn "gain" - which subtracts off this k percentage expected by chance. Note that when k is expressed as a percentage, the maximum gain is now 1-k and misaligned datasets can even have negative gain up to -k. With this calibration, randomly distributed data has an expected mutual-knn gain of zero across all values of k.

We've found these conventions useful in understanding how the alignment is behaving at various resolutions. For example, we can vary k to verify the mutual knn gain in case study one is maximized at k=9% and also characterize the degradation at finer and courser settings. (Figure 4)

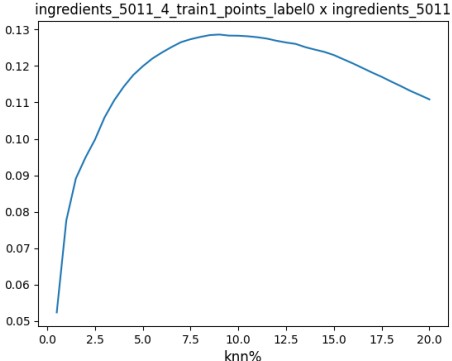

Figure 4: VISTA map mutual-knn gain in case study one with a maximum at k=9% (360)

## B   Appendix

Below are higher quality versions of the three VISTA maps covered in case studies. These VISTA maps are also available online at `http://got.drib.net/latents/` along with several others generated as part of this initial study.

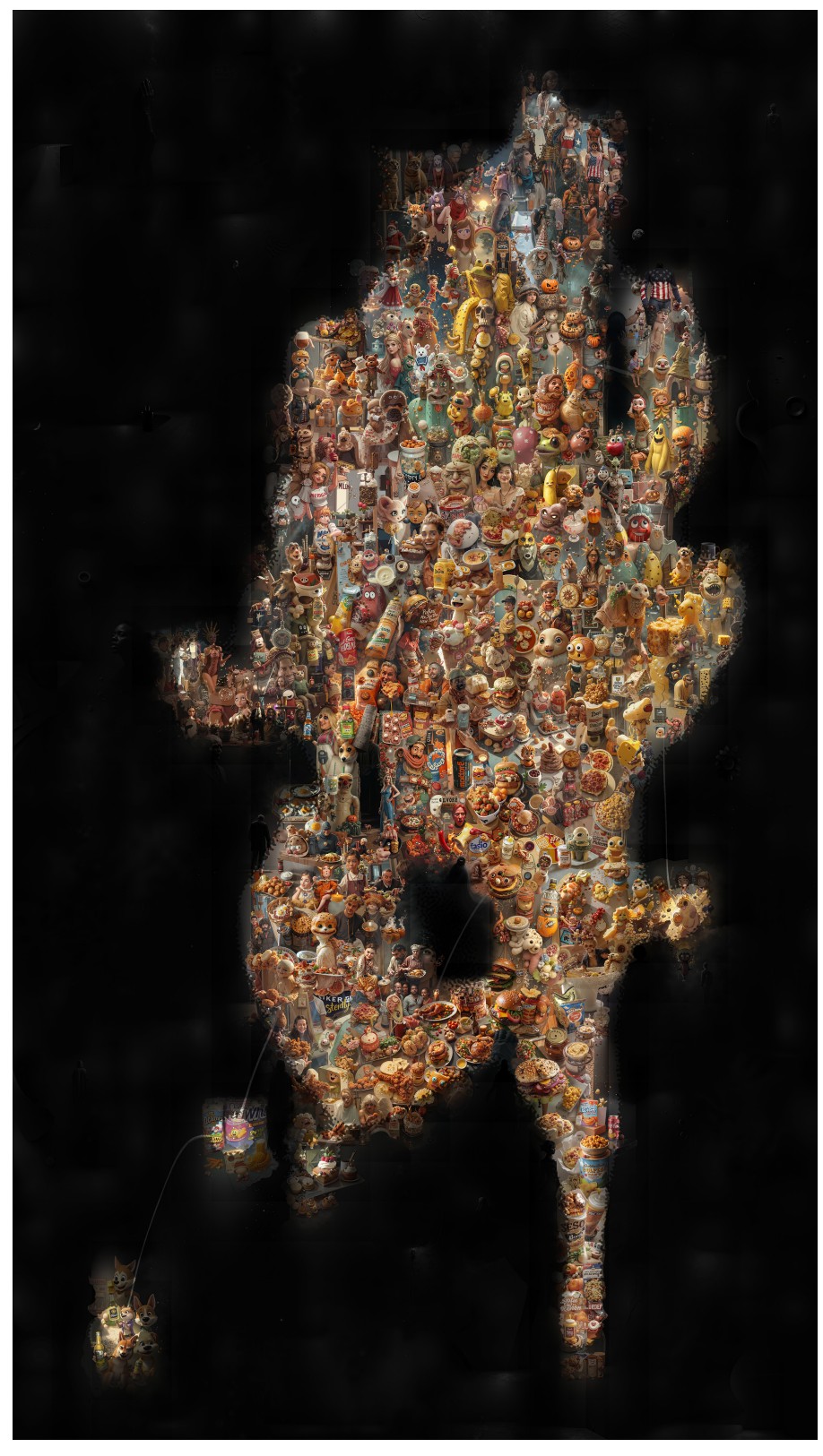

Figure 5: VISTA map for "ingredients" = latent gemma-2-2b/20-res-16k/5011. (web link)

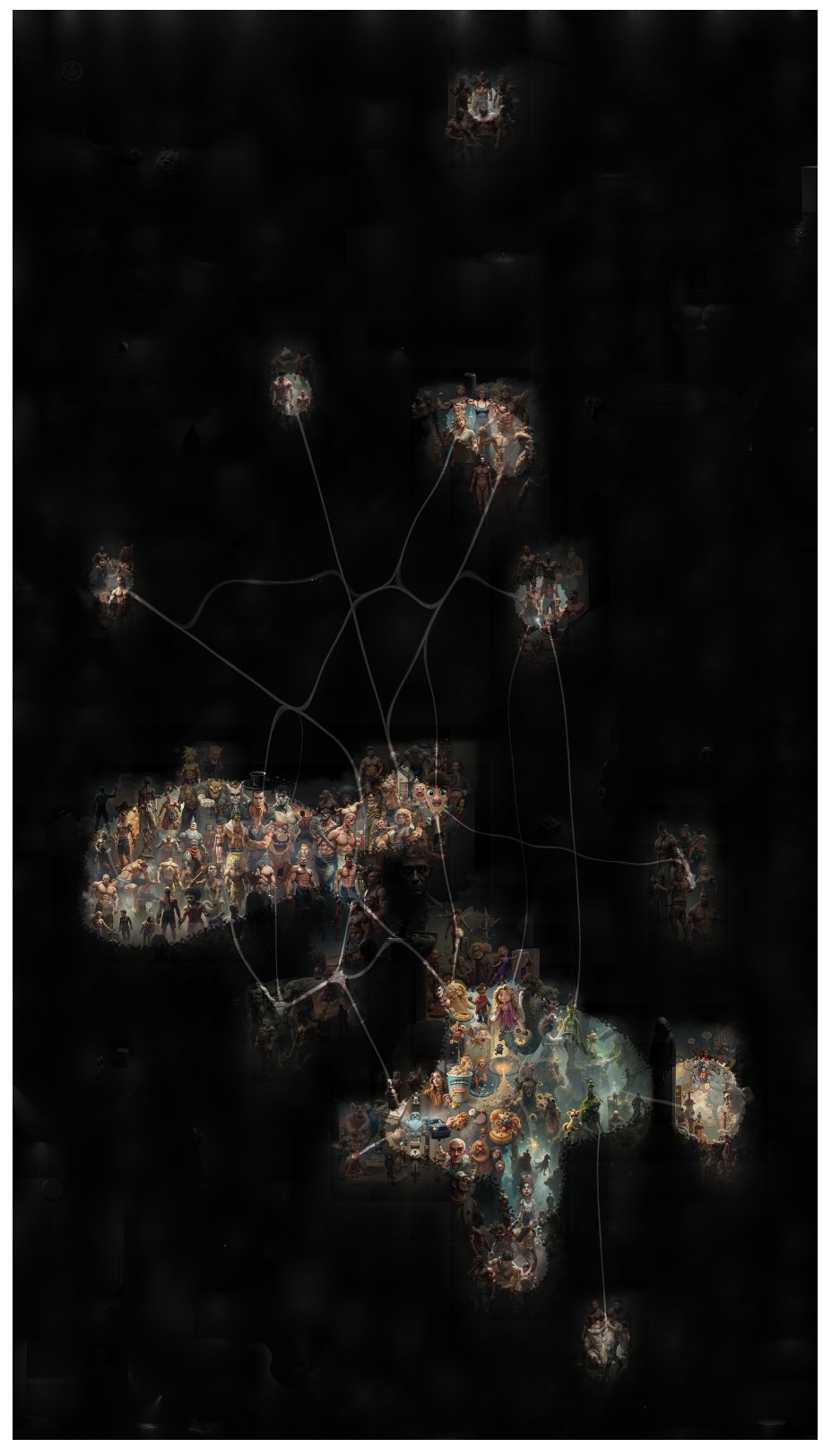

Figure 6: VISTA map for "muscle" = latent gemma-2-2b/20-res-16k/9745. (web link)

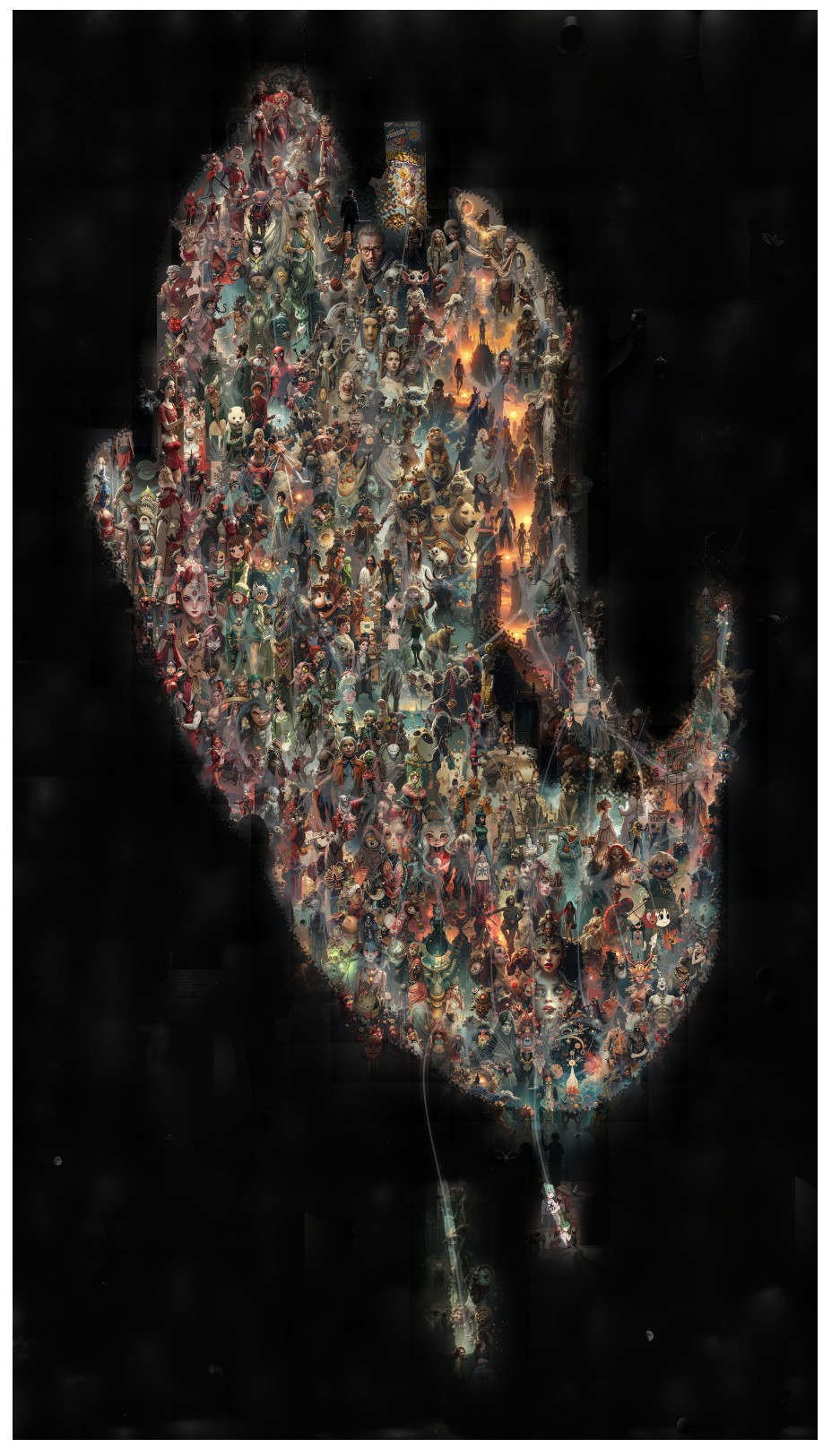

Figure 7: VISTA map for "indebted" = latent gemma-2-2b/20-res-16k/9220. (web link)

