# OpenReview forum: "VISTA: A Panoramic View of Neural Representations"
_NeurIPS.cc/2024/Workshop/UniReps — UniReps_

### Official Review · Reviewer_hHaa · 2024-10-02

**Rating:** 7
**Confidence:** 3

**Review:**

**Summary**

This paper introduces VISTA, a novel latent space visualization technique that provides a mosaic of summarizing clusters of input labels in a lower-dimensional representation. The authors apply this method to analyze the latent codes of sparse autoencoders, demonstrating its effectiveness in uncovering deeper meanings and associations that traditional automated techniques may overlook. VISTA aggregates labels of local neighborhoods in a smooth manner, facilitating the understanding of clusters containing multiple semantic meanings.

**Strengths and Weaknesses**

**Strengths**

1.  VISTA offers a significant improvement over standard latent space visualization techniques. While traditional methods often represent embeddings as point clouds labeled with input signals (e.g., Liu, Yang, et al., 2019), VISTA aggregates labels of local neighborhoods smoothly, enhancing the understanding of complex, multi-semantic clusters.

2. The method respects the uncertainty of latent space areas with no features by masking those regions. This approach preserves the topological properties of the latent manifolds learned by UMAP, ensuring a more accurate representation of the data structure.

**Weaknesses**

1. The "cartographic rendering" step, which subdivides the 2D representation of the latent space, may face challenges depending on the window size:
   - A large window size might over-generalize dense clusters, potentially obscuring semantic diversity.
   - A small window size could lead to excessive computational costs.

   A more thorough analysis of the tiling approach is needed to address these potential issues and optimize the balance between detail and efficiency.

2.  The reliance on generative models like MultiDiffusion introduces potential issues such as hallucinations or non-informative conditioning. These challenges could impact the data analysis capabilities of the method and should be addressed more explicitly.

3. While the paper focuses on text-based datasets, it lacks a comprehensive discussion on how VISTA might perform with other modalities such as images or audio, limiting its perceived versatility.

**Questions**

1. How do you anticipate VISTA will perform when applied to datasets in other modalities, such as images or audio? What modifications, if any, would be necessary to adapt the method for these different data types?

2. Could you provide more details on the "custom metric" used for UMAP? How was this metric developed, and what specific advantages does it offer over standard metrics in the context of your visualization technique?

3. Have you conducted any experiments or analysis on the optimal window size for the "cartographic rendering" step? How does the choice of window size affect the balance between computational efficiency and the preservation of semantic diversity in dense clusters?

**Limitations**

While the authors have briefly addressed the implications of their work, a more comprehensive analysis of the limitations would strengthen the paper. This could include:

1. An exploration of potential biases introduced by the visualization technique and how they might be mitigated.
2. An analysis of the computational resources required for VISTA compared to traditional visualization methods.
3. A consideration of how the method performs on datasets with highly imbalanced or sparse distributions in the latent space, which could affect the assumptions of UMAP.

Addressing these points would provide a more balanced view of VISTA's capabilities and potential areas for future improvement.

**Ethical concerns:** No

**Soundness:** 3 good

**Presentation:** 2 fair

**Contribution:** 3 fair

---

### Official Review · Reviewer_rzy1 · 2024-10-06
**Introduces a novel pipeline to visually explore LLMs' internal representation but lacks clarity**

**Rating:** 5
**Confidence:** 3

**Review:**

**Summary:** The paper introduces "VISTA", a novel pipeline to generate cartography of internal representation of LLMs for visual exploration and interpretation.

**Quality, clarity and originality:** The suggested pipeline is creative but it could’ve been great if the authors elaborated more on explaining the method of VISTA.

**Significance of the work:** The current best practice for understanding the meaning encoded in each latent is "automated interpretability", which provides suggested labels for each latent as it is entered into LLM. This work offers a unique approach to exploring the meaning of encoded latent visually.

**Strengths:**
- This paper touches on lots of topics that interest the workshop audience
- The paper introduces an interesting method to visually explore the internal representation of neural network models

**Weaknesses:**
- The paper is missing a detailed explanation of how to render cartographic renderings, which is the most important part of this work.
- This work lacks of ability to convince the reader why the approach that VISTA used is a better way to explore the internal representation.

---

### Official Review · Reviewer_SzDj · 2024-10-07
**Interpretability throguh 2D cartographic visualisation**

**Rating:** 5
**Confidence:** 3

**Review:**

The paper introduces a new framework (VISTA - Visualization of Internal States and Their Associations) for visualizing  (and potentially interpreting) neural representations of neural networks through what is referred to a "visual cartography" approach, i.e. the creation of a visual map that spatially represents the relationships between different elements of high-dimensional data.

The proposed pipeline goes as follows:
After projecting the high-dimensional representations into a 2D space using UMAP, VISTA identifies clusters of similar representations. These clusters represent regions in the 2D map where data points with similar latent features are grouped together. Then it uses **text-to-image diffusion models** to generate visual representations (images) for each of these clusters as part of a MultiDiffusion panorama. This is supposed to confer interpretability since data points with similar semantic meaning are going to be visualised nearby in the 2D space as images of similar content. The resulting 2D map is designed to be interactive, allowing users to zoom in and explore or probe specific regions.



They apply the proposed approach on sparse latents obtained from the Gemma2-2B sparse autoencoder, where current state of the art techniques for interpretability rely on large language models. Instead the proposed approach proposes to probe to neural representations of autoencoder -like models in the visual domain instead of the language domain


**Strengths:**
- The method provides visual insights into neural representations.
- The method can be applied across various neural models and datasets, making it adaptable to different research contexts.
-  The "cartographic" nature of the approach allows for the creation of interactive visual maps that make it easier to intuitively explore the structure of the data.


**Weaknesses:**
- The authors do not provide any discussion on the limitation of the method.
- I find that the underlying challenge of aligning visual results with semantic meaning is not convincingly addressed.
- Moreover I consider that the significance of the paper is a bit limited. While it addresses the important problem of interpreting high-dimensional neural representations, the practical impact of VISTA is unclear. The results presented in the paper, such as the identification of secondary triggers for certain latents, are interesting, but it is uncertain whether these insights could be reliably used to improve model interpretability. Furthermore, the reliance on visual inspection as a primary validation method makes it difficult to generalize the findings. VISTA might be of interest to researchers focused on interactive visualization, but as I understand it, it is unlikely to advance the state of the art in a substantial way, but I am happy to get convinced by the authors for the reverse.
- Also see questions.


**Questions:**

- The authors mention that they apply "a custom metric combining cosine distance in SAE space with the absolute difference along the latent axis". Can they provide a justification for this choice of metric?

- How do the authors ensure that the semantic labels and patterns identified by VISTA are not biased by the dataset or the text-to-image model used? Could they probably provide more details on the validation process?

- One question that I had while reading the paper, and I think it is the main point that needs clarification in order to make the acceptance of the paper reasonable, is what is the main insight that this method provides? As I understand similar contents are arranged nearby on the the 2D visualisation, as hopefully expected, but even ignoring the fact that the UMAP algorithm introduces some sort of randomness in the global arrangement of the clusters, I am not sure what sort of interpretability this method confers.

- The authors mention that to understand representational alignment they employ a mutual nearest neighbor metric, that however they do not describe further, so it is not clear to me what exactly this is.

- How would the method perform in cases where the latents do not represent objects but more abstract notions and concepts that are difficult to visualize?

- In the third example how do the authors conclude that the label of the Gemma score is incorrect and their visualization is correct? How can they ensure the reliability of VISTA's interpretations—whether the visual clusters identified actually reflect meaningful structure or if they, are influenced by biases in the text-to-image generation process. I am willing to believe them, but `i am not sure how one conclude which method is providing correct interpretation over the other.

- Considering that UMAP is used to generate the 2D projection into the collage-like representation, I would expect some systematic comparison on **i)** how different runs of UMAP with same hyperparameters influence resulting maps, and **ii)** how different values of the hyperparameters influence the resulting visualisations.

---

### Decision · Program_Chairs · 2024-10-10

**Decision:**

Accept

**Comment:**

In light of the positive reviewers' feedback and relevancy of the submission, we are pleased to accept this paper for presentation at UniReps 2024. We kindly ask the authors to incorporate the reviewers' suggestions and feedback in the final camera-ready version of the manuscript.